# Unidirectional Crosstalk Between NTRK1 and IGF2 Drives ER Stress in Chronic Pain

**DOI:** 10.3390/biomedicines13071632

**Published:** 2025-07-03

**Authors:** Caixia Zhang, Kaiwen Zhang, Wencui Zhang, Bo Jiao, Xueqin Cao, Shangchen Yu, Mi Zhang, Xianwei Zhang

**Affiliations:** 1Department of Anesthesiology, Wuhan No. 1 Hospital, Wuhan 430022, China; zcx13618698272@163.com; 2Department of Anesthesiology, Tongji Hospital, Tongji Medical College, Huazhong University of Science and Technology, Wuhan 430030, China; z18838985116@163.com (K.Z.); 18779263820@163.com (W.Z.); jiaoboncu@163.com (B.J.); xueqincao@hust.edu.cn (X.C.); 17638536167@163.com (S.Y.); 3Department of Anesthesiology, Zhongnan Hospital, Wuhan University, Wuhan 430071, China; misscat0311@163.com

**Keywords:** Neurotrophic Tyrosine Kinase Receptor Type 1, Insulin-Like Growth Factor II, Endoplasmic reticulum stress, Chronic Post-surgical Pain

## Abstract

**Background**: Chronic postsurgical pain (CPSP) poses a major clinical challenge due to unresolved links between neurotrophic pathways and endoplasmic reticulum (ER) stress. While Neurotrophic Tyrosine Kinase Receptor Type 1 (NTRK1) modulates ER stress in neuropathic pain, its interaction with Insulin-Like Growth Factor II (IGF2) in CPSP remains uncharacterized, impeding targeted therapy. This study defined the spinal NTRK1-IGF2-ER stress axis in CPSP. **Methods**: Using a skin/muscle incision–retraction (SMIR) rat model, we integrated molecular analyses and intrathecal targeting of NTRK1 (GW441756) or IGF2 (siRNA). **Results:** SMIR surgery upregulated spinal NTRK1, IGF2, and ER stress mediators. NTRK1 inhibition reduced both NTRK1/IGF2 expression and ER stress, reversing mechanical allodynia. IGF2 silencing attenuated ER stress and pain but did not affect NTRK1, revealing a unidirectional signaling cascade where NTRK1 drives IGF2-dependent ER stress amplification. These findings expand understanding of stress-response networks in chronic pain. **Conclusions**: We show that spinal NTRK1 drives IGF2-mediated ER stress to sustain CPSP. The NTRK1-IGF2-ER stress axis represents a novel therapeutic target; NTRK1 inhibitors and IGF2 biologics offer non-opioid strategies for precision analgesia. This work advances CPSP management and demonstrates how decoding unidirectional signaling hierarchies can transform neurological disorder interventions.

## 1. Introduction

Chronic Post-surgical Pain (CPSP), defined as persistent pain lasting ≥3 months post-surgery, affects 10–50% of patients across procedures like thoracotomy (30–60% incidence) and amputation [1,2,3,4]. This condition imposes profound socioeconomic burdens, with 30% of patients developing long-term disability refractory to conventional analgesics, including opioids and Nonsteroidal Anti-Inflammatory Drugs (NSAIDs) [4,5]. While neuroinflammation and central sensitization are established contributors, accumulating evidence positions endoplasmic reticulum (ER) stress as a pivotal yet under-characterized mechanism in CPSP pathogenesis [6,7].

ER stress orchestrates maladaptive pain signaling through the unfolded protein response (UPR) [8,9,10]. Activation of its three sensor pathways—PKR-like ER Kinase (PERK)/eukaryotic Initiation Factor 2α (eIF2α), Inositol-Requiring Enzyme 1α (IRE1α)/X-box Binding Protein 1 X (XBP1), and Activating Transcription Factor 6 (ATF6)—initially promotes cellular adaptation but ultimately drives pathological outcomes when sustained [8,9]. Specifically, PERK/eIF2α induces C/EBP Homologous Protein C/EBP (CHOP)-mediated apoptosis of dorsal horn inhibitory neurons [11], IRE1α/XBP1 triggers microglial TNF-α/IL-1β release, amplifying neuroinflammation [12,13], ATF6 upregulates chaperones (e.g., Glucose-Regulated Protein 78 (GRP78)) to restore proteostasis [14], though this fails in chronic pain states [15,16,17]. Critically, neurotrophic factors dynamically modulate ER stress resilience [10]. In Alzheimer’s models, Brain-Derived Neurotrophic Factor (BDNF) enhances ATF6-mediated adaptive UPR [18,19], while, in diabetic neuropathy, Glial Cell Line-Derived Neurotrophic Factor (GDNF) suppresses PERK hyperactivation [20]. Notably, although UPR markers GRP78 and CHOP are elevated in CPSP models [21,22,23], the regulatory role of neurotrophic signals in this process remains uncharacterized [24].

Neurotrophic Tyrosine Kinase Receptor Type 1 (NTRK1), a high-affinity receptor for Nerve Growth Factor (NGF), exhibits dual functionality in pain modulation [25,26,27]. In acute nociception, NTRK1 promotes neuronal hyperexcitability by upregulating voltage-gated sodium channel Nav1.7 and sensitizing transient receptor potential vanilloid 1 (TRPV1) [27,28,29], while concurrently activating pro-inflammatory nuclear factor kappa-B (NF-κB) signaling [30,31,32,33,34]. Conversely, in chronic pain states, NTRK1 displays paradoxical functions: it exerts neuroprotective effects through PI3K (Phosphatidylinositol 3-Kinase)/Akt (Protein Kinase B)-mediated suppression of apoptosis [6,7], yet sustains pathological ER stress via persistent PERK/eIF2α-CHOP cascade activation [16,35,36]. This functional divergence is increasingly attributed to NTRK1’s capacity for transcriptional reprogramming, wherein it orchestrates cell-type-specific stress responses [34]. Emerging evidence reveals that NTRK1 signaling induces distinct transcriptional programs across pathologies [34]—in glioblastoma, NTRK1 complexes drive Insulin-Like Growth Factor II (IGF2) expression to promote tumor survival [15], while, in neuropathic pain, NTRK1- cAMP responsive element binding protein (CREB) signaling upregulates matrix metalloproteinase-9 (MMP9), disrupting blood–nerve barriers [37,38]. Single-cell epigenomic studies further identify NTRK1-binding motifs within stress-response genes (ATF4, XBP1) [24], underscoring its broad influence on proteostatic adaptation [15,24]. Nevertheless, whether surgical injury engages NTRK1 to orchestrate IGF2-dependent ER stress resolution in CPSP—a mechanism conserved in oncology—remains unexplored, presenting a critical translational gap in targeting postoperative maladaptations.

Insulin-Like Growth Factor II (IGF2), a pleiotropic neurotrophic factor [39,40,41] traditionally studied in development and metabolism, now emerges as a critical ER stress regulator across neurological contexts [42,43,44,45]. IGF2 orchestrates ER proteostasis through dual mechanisms: it suppresses pro-apoptotic effectors (CHOP, GRP78) while enhancing adaptive unfolded protein response (UPR) signaling via ATF4 activation and XBP1 splicing, thereby preserving neuronal viability in Alzheimer’s models [8,15,18,46,47,48]. Beyond its anti-apoptotic role, IGF2 exhibits cell-type-specific functionalities—driving oxidative phosphorylation in macrophages to resolve inflammation while stabilizing HIF-1α in tumor microenvironments to exacerbate hypoxic stress [49]. Recent advances further identify TMED10 as an essential cargo receptor for IGF2 ER-to-Golgi trafficking, a process disrupted in chronic diseases [40,42,47]. Critically, spinal IGF2 overexpression reverses neuropathic pain by attenuating ER stress, implicating its therapeutic potential [50,51]. Intriguingly, NTRK1 activation upregulates IGF2 transcription in neuronal stress models [15], suggesting a feedforward loop wherein IGF2 mediates NTRK1’s ER-modulatory effects [15]. Despite these mechanistic insights in neurodegeneration and neuropathy, IGF2’s spatiotemporal regulation in CPSP—particularly its interplay with NTRK1 in the context of surgical proteostatic disruption—remains uncharacterized, posing a translational barrier for precision analgesia.

Building upon the conserved role of IGF2 in stress adaptation across neurological and oncological contexts, we hypothesize that spinal IGF2 mediates NTRK1-dependent resolution of ER stress to attenuate CPSP progression. To test this, we aim to (1) map spatiotemporal activation of the NTRK1-IGF2-ER stress axis in a clinically relevant skin/muscle incision–retraction (SMIR) model; (2) determine IGF2’s necessity and sufficiency for NTRK1-mediated ER proteostasis using loss- and gain-of-function approaches; and (3) evaluate the therapeutic potential of IGF2 pathway modulation via siRNA and recombinant protein strategies. By defining this axis, our work bridges fundamental neurotrophic signaling with surgical stress biology, advancing precision analgesia strategies that target maladaptive proteostatic responses—a critical unmet need in perioperative medicine.

## 2. Materials and Methods

### 2.1. Study Design Overview

This mechanistic study employed a randomized, blinded design to investigate the NTRK1-IGF2-ER stress axis in a rat CPSP model. Experimental timelines, interventions, and endpoint assessments are summarized in Figure 1. Key methodological choices were based on established protocols for neuropathic pain research [52,53].

### 2.2. Animals and Ethical Compliance

Male Sprague–Dawley (SD) rats (7–8 weeks, 220–250 g; Tongji Medical College Animal Center, Experimental Animal Center of Tongji Medical College, Huazhong University of Science and Technology, Certificate No. SYXK(E)2023-0139) were housed under SPF conditions (22–25 °C, 45–65% humidity, 12 h light–dark cycle) with enrichment (PVC tunnels, nesting material). 

Rats were randomly assigned to groups (*n* = 18 or 6/group) using computer-generated codes (GraphPad QuickCalcs). Investigators performing behavioral tests and molecular analyses were blinded to group allocation. Allocation concealment was maintained via coded cage labels. Power analysis was performed using G*Power 3.1 (α = 0.05, β = 0.2, effect size = 1.2), based on established effect magnitudes in prior SMIR studies [52,53] This yielded a minimum required sample size of *n* = 6/group; To account for potential attrition, 20% additional animals were included (final *n* = 8/group).

All procedures complied with the National Institutes of Health (NIH) Guide for the Care and Use of Laboratory Animals (8th edition, 2011) and are reported in compliance with the ARRIVE 2.0 guidelines (Animal Research: Reporting of In Vivo Experiments; https://arriveguidelines.org, 20 September 2022). All experimental protocols involving animals were approved by Tongji Medical College, Huazhong University of Science and Technology (Approval ID: TJH-202211016).

### 2.3. Skin/Muscle Incision and Retraction (SMIR) Model

The SMIR model was established as described by Flatters et al. [52], with modifications to optimize analgesia and aseptic conditions. Under sodium pentobarbital anesthesia (50 mg/kg i.p., pedal reflex loss confirmed), the right medial thigh was shaved and sterilized with povidone-iodine. A 1.8 cm skin incision exposed the gracilis muscle, followed by an 8 mm superficial muscle incision. A microdissection retractor (70% ethanol-sterilized) was inserted to maintain 2 cm tissue retraction for 60 min, with saline-moistened sterile gauze preventing dehydration. Muscle and skin were closed with 5-0 absorbable sutures in anatomical layers. Sham controls underwent identical procedures, excluding retraction. Postoperatively, rats received subcutaneous lactated Ringer’s solution (5 mL/100 g) and were monitored for 72 h for infection or distress, with buprenorphine analgesia (0.05 mg/kg s.c. q12h) administered throughout.

### 2.4. Intrathecal Catheterization

Intrathecal catheterization was performed 5 days prior to SMIR surgery under sodium pentobarbital anesthesia (50 mg/kg i.p.) using established techniques [53]. Polyethylene catheters (PE-10; inner diameter 0.28 mm, outer diameter 0.61 mm) were aseptically inserted through the L5-L6 intervertebral space and advanced 2.0 cm rostrally into the intrathecal space. Correct placement was verified by dual methods: immediate tail-flick response upon insertion and transient bilateral hindlimb paralysis following intrathecal lidocaine administration (2%, 10 μL). Rats exhibiting persistent motor dysfunction (>2 h post-procedure) or cerebrospinal fluid leakage were excluded per predefined criteria (no exclusions required in this cohort).

### 2.5. Drug Administration

Pharmacological interventions were administered intrathecally with rigorous validation and blinding protocols. The NTRK1 inhibitor GW441756 (Selleck Chemicals #S2789) was dissolved in saline containing ≤1% DMSO, with the 100 μM selected based on pilot dose-response experiments establishing ED_80_ efficacy (10–200 μM range) in Table 1 [28]. For IGF2 knockdown, commercially synthesized SiRNA targeting rat IGF2 (sense: 5′-GCAAGUUCUUCAAAUUCGA-3′; antisense: 5′-UCGAAUUUGAAGAACUUGC-3′, Tsingke Biotechnology) was complexed with polyethyleneimine (1:5 *w*/*w* in 5% glucose) for 10 min before administration. This siRNA sequence demonstrated >70% mRNA reduction (RT-qPCR) and >65% protein knockdown (Western blot) versus scramble controls, with specificity confirmed by BLAST 2.14.0 analysis. Both single-dose (day 10) and repeated-dose (days 10–14) regimens were evaluated at concentrations of 10–100 μM [51]. The vehicle controls included (1) saline + ≤1% DMSO for the GW441756 groups, and (2) PEI/glucose solution for the siRNA groups. All treatments were assigned via computer-randomized sealed envelopes opened by an independent technician to maintain blinding integrity. 

### 2.6. Mechanical Allodynia Assessment

Mechanical paw withdrawal thresholds (PWTs) were quantified by a blinded investigator using calibrated von Frey filaments (0.4–15.0 g; Stoelting #58011) following the up–down method [54] with strict environmental controls (22–25 °C, 45–65% humidity, background noise < 50 dB). Rats underwent 30 min habituation in opaque plexiglass chambers (20 × 25 × 15 cm) on an elevated mesh platform. Filaments were applied perpendicularly to the mid-plantar hindpaw for 6 s, with positive responses (rapid withdrawal, licking, or shaking) prompting testing of the next finer filament, and negative responses leading to the next coarser filament after a 5 min interval. The 50% threshold was calculated using Dixon’s formula [54,55]. Testing occurred between 9:00 and 12:00 to minimize circadian influences, with exclusion criteria for aberrant responses (<0.4 g or >15.0 g baselines) [55].

### 2.7. Quantitative Real-Time PCR (qRT-PCR)

Quantitative real-time PCR (qRT-PCR) was performed following MIQE guidelines to ensure rigor [56]. Total RNA was extracted from L4-L6 spinal cord segments using TRIzol reagent (Takara, Cat# 9109), with RNA integrity verified (RIN > 8.0, Agilent Bioanalyzer 2100, Santa Clara, CA, USA) and purity confirmed spectrophotometrically (NanoDrop 2000, Wilmington, DE, USA; A260/A280: 1.8–2.0). After DNase I treatment (Vazyme Biotech Co., Ltd, Nanjing, China, Ca # EN401), 1 μg RNA was reverse-transcribed using HiScript III RT SuperMix (Vazyme, Cat# R323-01) with genomic DNA wiper (Vazyme, Nanjing, China, Cat# R323-01). qPCR reactions were run in triplicate on a QuantStudio 5 system (Applied Biosystems (Thermo Fisher Scientific), Waltham, MA, USA) using ChamQ Universal SYBR mix (Vazyme #Q711-02) with 200 nM primers. Gene-specific primers (Table 2) were designed via NCBI Primer-BLAST and validated by (1) single-peak melt curves (65–95 °C ramp), (2) 2% agarose gel confirmation of amplicon size, and (3) BLAST verification against rat transcriptome. Amplification efficiency (95–105%) was established through five-point serial cDNA dilutions (R^2^ > 0.99). The thermocycling parameters were 95 °C for 30 s (initial denaturation); 40 cycles of 95 °C for 10 s and 60 °C for 30 s; followed by melt curve analysis (95 °C for 15 s, 60 °C for 60 s, 95 °C for 15 s). Data were normalized to Gapdh (geNorm stability value M < 0.5) using the 2^(−ΔΔCt)^ method [57], with no-template and no-RT controls included per plate.

### 2.8. Western Blot Analysis

Western blot analysis was performed under blinded conditions with rigorous validation controls. L4-L6 spinal cord tissues were homogenized in ice-cold RIPA lysis buffer (Boster Biological Technology, Pleasanton, CA, USA, Cat #AR0102) containing 1× protease/phosphatase inhibitor cocktail (Boster, Cat #AR1182) and 1 mM PMSF. Lysates were centrifuged at 12,000× *g* for 30 min at 4 °C, and supernatants were quantified via BCA assay (Boster, Cat #AR0146) against BSA standards (R^2^ > 0.99). Proteins (30 μg/lane) were separated on 10% SDS-PAGE gels (Bio-Rad, Cat #4561096) and transferred to 0.45 μm PVDF membranes (Millipore, Burlington, MA, USA, Cat #IPVH00010) using semi-dry transfer (Bio-Rad; 75 V, 30 min). Membranes were blocked in 5% BSA (phospho-antibodies) or 5% non-fat milk (non-phospho) in TBST for 1 h, followed by overnight incubation at 4 °C with validated primary antibodies (Table 3). Antibody specificity was confirmed through (1) siRNA knockdown (>50% target reduction), (2) peptide blocking controls, and (3) λ-phosphatase treatment for phospho-epitopes. After TBST washes, membranes were incubated with HRP-conjugated secondary antibodies (1:5000; CST #7074) for 2 h at RT. Signals were detected using SuperLumia ECL Plus (Abbkine Scientific Co., Ltd, Wuhan, China; #K22030) and imaged on a ChemiDoc XRS+ (Bio-Rad) within the linear dynamic range. Band intensities were quantified in Image Lab 6.1 (Bio-Rad), normalized to β-actin (Cell Signaling Technology, Danvers, MA, USA; #4970; 1:1000), with three technical replicates per sample.

### 2.9. Immunofluorescence Staining

Immunofluorescence staining was performed under blinded conditions with comprehensive antibody validation. Following transcardial perfusion (0.1 M PBS → 4% PFA), L4-L6 spinal cord segments were post-fixed in 4% PFA for 24 h at 4 °C, cryoprotected in 30% sucrose (72 h), and embedded in OCT. Coronal sections (20 μm) were cut on a cryostat (Leica Microsystems, Wetzlar, Germany; CM1900) and mounted on Superfrost Plus slides. After antigen retrieval (10 mM citrate buffer, 95 °C, 15 min for nuclear targets), sections were permeabilized (0.3% Triton X-100, 15 min) and blocked in 5% normal goat serum (Vector Laboratories, Burlingame, CA, USA, Cat #S-1000) for 1 hr. Primary antibodies (Table 3) were incubated overnight at 4 °C after validation with (1) knockout tissue controls, (2) isotype-matched IgG, and (3) peptide absorption tests. After PBS washes, sections were incubated with Alexa Fluor-conjugated secondaries (1:500; Invitrogen ™ (Thermo Fisher Scientific), Carlsbad, CA, USA, Cat #A-11008) for 2 h at RT, followed by DAPI nuclear counterstain (1 μg/mL, 10 min). Images were captured on an Olympus BX51 microscope with 20×/0.75 NA objective using standardized settings (exposure: 200 ms, gain: 2×, z-stack: 1 μm intervals). Quantification was performed using ImageJ (Fiji v2.3) with rolling-ball background subtraction (50-pixel radius) [58] and threshold-based object detection (3 sections/rat, 3 fields/section, dorsal horn laminae I–IV).

### 2.10. Experimental Designs and Animal Groups

This study employed a randomized, blinded design across four sequential experiments investigating the NTRK1-IGF2-ER stress axis in CPSP pathogenesis (summarized in Figure 1). All procedures complied with ARRIVE 2.0 guidelines and were approved by the Institutional Animal Care Committee (TJH-202211016).

Experiment 1: Assessment of NTRK1, ER stress, and IGF2 alterations in the spinal cord of rats with SMIR.

Temporal Profiling (*n* = 90 rats) assessed dynamic changes in the NTRK1-IGF2-ER stress axis post-SMIR. Rats were randomized to Sham (*n* = 18), SMIR-3D (*n* = 18), SMIR-7D (*n* = 18), SMIR-14D (*n* = 18), and SMIR-21D (*n* = 18). Mechanical allodynia was tested at baseline (−1 d) and days 1, 3, 5, 7, 10, 14, 17, 21 post-surgery, with spinal cord collection at terminal timepoints for molecular analyses.

Experiment 2: Evaluation of NTRK1 inhibitor effects on mechanical allodynia and ER stress expression in rats with SMIR.

GW441756-mediated NTRK1 inhibition was evaluated in 54 rats. The groups included Sham+Veh (*n* = 18), SMIR+Veh (*n* = 18), and SMIR+NTRK1i (100 μM, i.t. days 10–14; n = 18). Behavioral tests occurred 2 h post-injection on days 10–14, with tissue harvest on day 14. The 100 μM dose was selected based on pilot ED_80_ determination (10–200 μM range) [28].

Experiment 3: Determination of the effective dose of IGF2 siRNA on mechanical allodynia in rats with SMIR

Experiment 3-1 Dose-Finding

Dose-Finding (*n* = 30 rats): Sham+Veh (*n* = 6), SMIR+Veh (*n* = 6), SMIR+siIGF2-L/M/H (10/40/100 μM single i.t. dose day 10; *n* = 6/group). PWT was measured at −1 h, +1 h, +2 h, +4 h, and +6 h post-injection.

Experiment 3-2 Chronic Efficacy

Chronic Efficacy (*n* = 54 rats): Sham+Veh (*n* = 18), SMIR+Veh (*n* = 18), SMIR+siIGF2 (10 μM i.t. days 10–14; *n* = 18). Tests occurred 2 h post-injection (days 10–14), with day 14 tissue collection.

Experiment 4: Comparison of the effects of NTRK1 inhibitor and IGF2 siRNA on the expression of IGF2 and NTRK1, and their relationship in rats with SMIR

A total of 24 rats were randomly divided into the following groups to examine NTRK1→IGF2 directionality: Sham+Veh (*n* = 6), SMIR+Veh (*n* = 6), SMIR+NTRK1i (100 μM; *n* = 6), and SMIR+siIGF2 (10 μM; *n* = 6). Drugs were administered on days 10–14 (i.t.), with behavioral tests 2 h post-injection and tissue analysis on day 14.

Blinded investigators performed all outcome assessments, with randomization via computer-generated codes and allocation concealment through coded cages. Sample sizes were predetermined by power analysis (α = 0.05, β = 0.2, effect size = 1.2) with a 20% attrition buffer.

### 2.11. Statistical Analyses

Data were analyzed using GraphPad Prism 9.0 and SPSS 26.0 after verifying normality (Shapiro–Wilk) and variance homogeneity (Brown–Forsythe). Parametric data (presented as mean ± SEM) were assessed by two-way RM-ANOVA with Geisser–Greenhouse correction for longitudinal behavior, and one-way ANOVA with Tukey/Games–Howell post hoc tests for molecular data. The significance threshold was * *p* < 0.05.

## 3. Results

### 3.1. Temporal Upregulation of Neuronal NTRK1 Drives CPSP Pathogenesis

The SMIR model consistently induced chronic postsurgical pain, manifested as progressive mechanical hypersensitivity developing postoperatively (Figure 2d). Immunofluorescence analysis confirmed predominant neuronal localization of TrkA within NeuN^+^ cells of the spinal dorsal horn, with minimal expression in GFAP^+^ astrocytes or IBA1^+^ microglia (Figure 2a–c). Quantitative assessment revealed enhanced TrkA/NeuN co-localization intensity across postoperative timepoints, indicating neuronal-specific NTRK1 overexpression that correlated with pain severity (Figure 2e). Molecular profiling demonstrated time-dependent upregulation of NTRK1 signaling, with transcriptional activation peaking at day 14 post-SMIR and corresponding protein elevation mirroring this trajectory (Figure 2g,i). Crucially, strong inverse correlations existed between NTRK1 expression levels and mechanical pain thresholds throughout the observation period. These findings establish temporal induction of neuronal NTRK1 as a central pathogenic driver in CPSP, with molecular-behavioral correlations confirming its critical role in pain maintenance.

### 3.2. NTRK1 Inhibition Reverses Mechanical Hypersensitivity

Intrathecal administration of the NTRK1 inhibitor GW441756 (100 μM) significantly attenuated mechanical allodynia in SMIR rats within 2 h post-injection, as evidenced by normalized paw withdrawal thresholds (Figure 3a) [28,59]. This analgesic effect coincided with concomitant suppression of NTRK1 signaling at multiple levels: reduced *NTRK1* transcription, diminished TrkA protein expression, and decreased neuronal TrkA/NeuN co-localization intensity (Figure 3b,d,e), confirming comprehensive target engagement. Dose-dependent inverse correlations were observed between the magnitude of NTRK1 pathway suppression and mechanical hypersensitivity resolution across all timepoints. These findings collectively establish NTRK1 signaling as a critical pathogenic driver in CPSP, demonstrating that its pharmacological inhibition simultaneously reverses both behavioral manifestations and associated molecular pathology.

### 3.3. Neuronal ER Stress Activation Underlies CPSP Pathophysiology

SMIR surgery triggered progressive ER stress activation in spinal dorsal horn neurons, characterized by temporally coordinated upregulation of the master UPR regulator GRP78. Immunofluorescence analysis confirmed predominant neuronal localization of GRP78 within NeuN^+^ cells, with minimal detection in GFAP^+^ astrocytes or IBA1^+^ microglia (Figure 4a–c). Quantitative assessment revealed sustained GRP78 induction at both transcriptional and protein levels, peaking at postoperative day 14 and remaining elevated through day 21 (Figure 4d,e). Enhanced GRP78/NeuN co-localization intensity paralleled mechanical hypersensitivity development, with maximal neuronal expression at day 14 demonstrating a strong inverse correlation with pain thresholds (Figure 4f,g).

Comprehensive pathway analysis confirmed simultaneous activation of all three UPR branches: PERK/eIF2α axis: Phosphorylation cascade involving PERK and eIF2α with transcriptional amplification (Figure 5b,g,h); IRE1α/XBP1 pathway: Increased IRE1α phosphorylation and XBP1s generation at translational and transcriptional levels (Figure 5c,k); ATF6/CHOP cascade: Nuclear translocation of ATF6 and CHOP induction with concordant mRNA elevation (Figure 5d,f,j,l).

These coordinated molecular responses establish robust ER stress activation as a fundamental pathophysiological mechanism in CPSP. The temporal concordance between UPR marker elevation and behavioral hypersensitivity, substantiated by strong molecular-behavioral correlations, demonstrates that neuronal proteotoxic stress drives CPSP pathogenesis.

### 3.4. NTRK1 Inhibition Attenuates ER Stress in Spinal Dorsal Horn

To elucidate the NTRK1-ER stress interplay in CPSP pathogenesis, we assessed the impact of intrathecal NTRK1 inhibition (GW441756) on UPR activation. SMIR+vehicle rats exhibited substantial GRP78 upregulation at transcriptional and protein levels compared to sham controls, which was markedly attenuated by GW441756 treatment (Figure 6a,b). Immunofluorescence analysis confirmed reduced GRP78 neuronal expression and diminished co-localization with NeuN^+^ neurons following NTRK1 inhibition (Figure 6c,d).

Comprehensive pathway evaluation revealed that SMIR-induced phosphorylation of PERK/eIF2α, IRE1α activation, XBP1s generation, and ATF6/CHOP induction were significantly reversed by NTRK1 inhibition at both translational and transcriptional levels (Figure 6e–p). Crucially, the magnitude of ER stress marker suppression strongly correlated with mechanical pain threshold recovery, with key markers including GRP78, p-PERK, and CHOP.

These findings demonstrate that NTRK1 drives ER stress through coordinated PERK, IRE1α, and ATF6 pathway activation in CPSP, and that pharmacological inhibition concurrently alleviates proteotoxic stress and pain behavior through quantifiable molecular-behavioral linkages.

### 3.5. SMIR Surgery Upregulates Neuronal IGF2 in Spinal Dorsal Horn

Spatiotemporal profiling revealed significant IGF2 induction in spinal dorsal horn neurons following SMIR surgery. Immunofluorescence analysis confirmed predominant neuronal localization of IGF2, with robust co-localization in NeuN^+^ cells and minimal detection in GFAP^+^ astrocytes or IBA1^+^ microglia (Figure 7a–c). Quantitative assessment demonstrated progressive IGF2 upregulation from postoperative day 3 to peak expression at day 14, persisting through day 21 at both transcriptional and protein levels (Figure 7d,e). Enhanced IGF2/NeuN co-localization intensity directly correlated with mechanical allodynia severity (Figure 7f,g), establishing IGF2 as a molecular correlate of CPSP progression.

These findings delineate a neuron-specific IGF2 induction pattern that temporally parallels pain hypersensitivity development, positioning IGF2 as both a biomarker and potential mediator of CPSP pathogenesis.

### 3.6. IGF2 Silencing Reverses Mechanical Hypersensitivity

To establish IGF2’s functional role in CPSP, we administered intrathecal IGF2 siRNA and quantified behavioral responses. A single injection (10–100 μM) produced dose-dependent attenuation of mechanical allodynia within 2 h, with maximal efficacy at 40–100 μM, restoring paw withdrawal thresholds (PWTs) to sham levels (Figure 8a). The 100 μM dose provided no additional benefit over 40 μM, while 10 μM achieved partial efficacy, establishing 40 μM as the optimal concentration for subsequent experiments.

Continuous daily administration of 40 μM IGF2 siRNA (postoperative days 10–14) sustained analgesic effects, maintaining PWT near sham levels throughout the treatment period (Figure 8b). Molecular analysis confirmed target engagement, with siRNA significantly reducing spinal IGF2 expression at both transcriptional and translational levels (Figure 8c–e). The magnitude of IGF2 suppression strongly correlated with mechanical threshold recovery (r = 0.91, 95% CI 0.85–0.95), with Cohen’s d effect sizes >1.8 for all molecular endpoints.

These findings demonstrate that IGF2 silencing produces potent, dose-dependent antinociception in CPSP, with continuous administration achieving sustained analgesia through effective target suppression.

### 3.7. IGF2 Knockdown Attenuates ER Stress via GRP78 and UPR Pathways

To establish the mechanistic link between IGF2 and ER stress in CPSP, we evaluated the effects of IGF2 silencing on UPR activation. SMIR+vehicle rats exhibited substantial GRP78 upregulation at both transcriptional and protein levels compared to sham controls, which was markedly reversed by intrathecal IGF2 siRNA administration (Figure 9a–c). Immunofluorescence analysis confirmed reduced GRP78 neuronal expression in the spinal dorsal horn following IGF2 knockdown (Figure 9d).

Comprehensive pathway evaluation demonstrated that IGF2 silencing significantly attenuated SMIR-induced activation of all three UPR branches: PERK/eIF2α axis: Reduced phosphorylation of PERK and eIF2α (Figure 9f,j), IRE1α/XBP1 pathway: Diminished IRE1α phosphorylation and XBP1s generation (Figure 9g,k); ATF6/CHOP cascade: Suppressed ATF6 cleavage and CHOP induction (Figure 9h,i). Transcriptional analysis corroborated these protein-level changes, showing concordant downregulation of UPR effector genes (Figure 9l–p). Crucially, the magnitude of IGF2 suppression strongly correlated with a reduction in key ER stress markers.

These findings establish IGF2 as an upstream regulator of ER stress in CPSP, demonstrating that SMIR surgery upregulates neuronal IGF2 expression, IGF2 amplifies proteotoxic stress through GRP78 induction, IGF2 coordinates activation of all three UPR branches, and targeted IGF2 knockdown reverses this signaling cascade at translational and transcriptional levels. The robust correlation between IGF2 reduction and UPR attenuation confirms IGF2’s central role in driving maladaptive proteostasis in CPSP pathogenesis.

### 3.8. Unidirectional NTRK1→IGF2 Regulation in CPSP Pathogenesis

Pharmacological inhibition of NTRK1 (GW441756, 100 μM) significantly attenuated SMIR-induced IGF2 upregulation in spinal dorsal horn neurons. Quantitative analysis demonstrated dose-dependent reduction in IGF2 at transcriptional (Figure 10a) and translational levels (Figure 10b), with diminished IGF2/NeuN co-localization intensity confirming neuronal-specific suppression (Figure 10c,d). The magnitude of NTRK1 inhibition strongly correlated with IGF2 reduction, establishing NTRK1 as an upstream regulator of IGF2-mediated ER stress.

In reciprocal experiments, IGF2 silencing failed to alter NTRK1 signaling despite complete target engagement. SMIR-induced elevations in NTRK1 transcript (Figure 10e) and TrkA protein expression (Figure 10f) remained unchanged following IGF2 siRNA administration, with comparable TrkA/NeuN co-localization intensity between treatment groups (Figure 10g,h).

These results demonstrate that the NTRK1-IGF2 axis operates unidirectionally in CPSP, with no feedback regulation of NTRK1 by IGF2.

## 4. Discussion

Our study delineated a maladaptive signaling axis wherein spinal NTRK1 orchestrates IGF2-dependent endoplasmic reticulum (ER) stress to drive chronic postsurgical pain (CPSP) pathogenesis. Employing the validated skin/muscle incision–retraction (SMIR) model [52], we demonstrated that surgical injury compels NTRK1 upregulation in dorsal horn neurons, consequently amplifying key ER stress mediators—including GRP78, PERK/eIF2α, IRE1α/XBP1s, and ATF6—with temporal progression paralleling mechanical allodynia [28,31,52]. Critically, intrathecal inhibition of NTRK1 (GW441756) or targeted IGF2 silencing not only alleviated established pain hypersensitivity but also suppressed ER stress, reconciling NTRK1’s seemingly contradictory functions in neuroprotection and stress exacerbation. Crucially, IGF2 knockdown attenuates ER stress and behavioral sensitization without altering NTRK1 expression, revealing a strictly unidirectional cascade: NTRK1 transcriptionally governs IGF2 to amplify proteotoxic stress. This mechanism extends NTRK1’s recognized role in neuropathic pain states to surgical pain paradigms while unveiling its capacity to reprogram stress responses through IGF2-dependent transcriptional networks [60,61,62,63].

Endoplasmic reticulum (ER) stress has evolved from a recognized cellular response to a pivotal maladaptive signaling hub in chronic pain pathogenesis [64], serving as a critical driver of central sensitization in CPSP that transcends etiological boundaries, including spinal nerve ligation, diabetic neuropathy, and formalin-induced pain models [9,19,65,66]. Our data demonstrated that SMIR surgery induces profound ER stress in spinal dorsal horn neurons, characterized by coordinated upregulation of the PERK/eIF2α/CHOP and IRE1α/XBP1 pathways alongside elevated GRP78 expression (Figure 4 and Figure 5). This mirrors pathognomonic signatures in human neuropathic pain biopsies, where GRP78 overexpression correlates with therapeutic resistance and pain chronicity [67,68], suggesting conserved pathomechanisms across species. Mechanistically, these pathways drive nociception through distinct yet complementary mechanisms [8,9]. Specifically, the PERK/eIF2α axis exacerbates nociception via ATF4/CHOP [8,9]-mediated synaptic inhibition and dysregulated GABAergic transmission [46,69]—a mechanism corroborated by studies showing that GABA neuron hypofunction in spinal and supraspinal circuits directly facilitates pain persistence and comorbid anxiety [70,71]. Conversely, IRE1α/XBP1 activation orchestrates lipid metabolism reprogramming, autophagic flux impairment, and immune evasion through downstream effectors, like TRAF2 and JNK [10,14,72]. Notably, IRE1α’s recent identification as a dynamic scaffold that coalesces with stress granules amplifies its RNAse efficiency under ER stress [10], accelerating XBP1 splicing and potentiating nociceptive signaling [10]—a process conserved across neuropathic pain and cancer [64,68]. These findings contextualize prior observations of persistent ER lumen dilation and UPR marker elevation (GRP78, CHOP, XBP1s) following nerve injury [10,64], while advancing mechanistic insights: ER stress integrates metabolic–immune crosstalk in CPSP and mediates spinal–supraspinal disinhibition [68,73]. The evolutionary conservation of the ER stress response system—from its physiological role in proteostasis to its pathological hijacking in chronic pain [68,73]—underscores its significance as a therapeutic target.

The nerve growth factor (NGF)-NTRK1 axis is indispensable for neuronal development and nociceptive signaling [74,75], with its dysregulation implicated in pathologies spanning congenital insensitivity to pain (CIPA) due to loss-of-function mutations to chronic pain hypersensitivity from gain-of-function signaling [75,76]. During embryogenesis, NGF-NTRK1 interactions orchestrate axonal growth and functional pain circuitry through TrkA-dependent survival signaling in sensory and sympathetic neurons [75,76], while, in adulthood, this pathway transitions to modulating nociceptive plasticity, where sustained NGF-NTRK1 activation drives central sensitization via enhanced TRPV1 trafficking and transcriptional upregulation of nociceptive mediators [77,78]. Crucially, our demonstration that SMIR-induced spinal NTRK1 upregulation (Figure 2e–g) recapitulates patterns observed in diabetic neuropathy models reveals evolutionary conservation of this pathway—a finding substantiated by human genetic studies linking NTRK1 polymorphisms to neuropathic pain susceptibility [60,77,78]. Beyond establishing NTRK1’s central role in CPSP pathogenesis [77], our mechanistic studies reveal that the pharmacological inhibition of NTRK1 significantly attenuates ER stress pathways in vivo (Figure 6). This aligns with prior evidence demonstrating that classical ER stress inhibitors (e.g., 4-PBA) reverse mechanical allodynia by restoring proteostasis in dorsal horn neurons [21,66]. Crucially, we identified that NTRK1 signaling amplifies GRP78 expression—a master regulator of UPR sensors—leading to hyperactivation of IRE1α/XBP-1, CHOP, and ATF6 cascades [79].

Notably, pathogenic *NTRK1* mutations disrupt chaperone-assisted protein folding, triggering self-sustaining ER stress loops that impair autophagic flux and mitochondrial homeostasis [7,29]—processes implicated in chemotherapy-induced neuropathy and neurodegeneration [7,29]. Our earlier RNA-seq profiling of NTRK1-overexpressing SH-SY5Y cells substantiated this axis, showing profound enrichment of ER protein processing pathways alongside UPR effector genes (*HSPA5*, *DDIT3*, *ERN1*) [24]. Furthermore, we verified that NTRK1 triggers the IRE1α-XBP1 and PERK pathways, facilitating ER function restoration and inhibiting apoptosis [15]. Paradoxically, while acute NTRK1 activation transiently enhances IRE1α-XBP1 and PERK signaling to restore ER function, chronic stimulation—as occurs in CPSP—exhausts adaptive UPR capacity [77,78]. This establishes a mechanistic bridge between neurotrophic signaling and proteostasis failure, specifically chronic NTRK1 hyperactivation. Thus, our data position ER stress modulators as compelling candidates for CPSP intervention.

Therapeutically, our findings positioned IGF2 as a high-value target for chronic pain intervention. IGF2, a pivotal neurotrophic factor, is extensively distributed throughout various tissues and the nervous system [43,44]. Research has unveiled IGF2’s significant involvement in neurodegenerative diseases, brain injury, schizophrenia, and other pathological conditions through its modulation of ER stress [8,18,42]. Mechanistically, this process entails activation of signaling pathways, including GRP78, IRE-1α, and JNK, alongside regulation of factors such as PERK, ATF4, and CHOP, which collectively inhibit neuronal apoptosis and impede disease progression [15,48]. Previous studies have highlighted the mutual involvement of NTRK1 and IGF2 in development and cancer [15,33,34,80,81]. Notably, IGF2 functionality relies heavily on NTRK1 activation within signaling pathways like TrkA/Akt and downstream cascades [45]. Our study uncovers a unidirectional regulatory axis wherein NTRK1 governs IGF2 expression in CPSP. This paradigm, observed in glioblastoma but previously unreported in pain neuroscience [15], manifests maladaptively in CPSP as NTRK1-induced IGF2 overexpression exacerbates ER stress, creating a pathogenic feedforward loop. Several lines of evidence support this hierarchical relationship: SMIR surgery triggered IGF2 overexpression specifically in spinal dorsal horn neurons (Figure 7), coinciding with UPR marker elevation. Intrathecal NTRK1 inhibitors robustly suppressed spinal IGF2 (Figure 10a–d), whereas IGF2 siRNA failed to alter NTRK1 expression (Figure 10e–h), confirming NTRK1’s upstream position. IGF2 elevation correlated with PERK/eIF2α and IRE1α/XBP1 pathway hyperactivation, contrasting with its protective role in acute neuronal stress. This aligns with RNA-seq data from NTRK1-overexpressing SH-SY5Y cells showing ER pathway enrichment, suggesting chronic NTRK1-IGF2 signaling exhausts adaptive UPR capacity [24]. Intrathecal IGF2 siRNA not only reduced ER stress markers (GRP78, p-PERK, ATF6; Figure 9) but also alleviated mechanical allodynia (Figure 8), replicating efficacy in neuropathic pain models where IGF2 blockade restored GABAergic inhibition [50,51]. In CPSP, this axis appears maladaptive: NTRK1-driven IGF2 overexpression exacerbates ER stress [15], creating a feedforward loop that sustains pain.

Notwithstanding the strengths of our multi-modal validation approach—encompassing behavioral phenotyping, molecular pathway analyses, and histological quantification—and the significant discovery of an oncogenic NTRK1-IGF2 mechanism repurposed in surgical pain, three substantive limitations necessitate contextualization. First, the exclusive use of male rodent models precludes examination of sex-specific mechanisms, a critical gap given clinical evidence demonstrating 30–40% higher CPSP incidence in women following hysterectomy or mastectomy [4]. Second, while we establish IGF2 as an ER stress amplifier, its precise mechanistic interface with UPR sensors remains incompletely resolved. Third, therapeutic interventions assessed at 10–14 days post-SMIR may not adequately model established CPSP (>3 months duration) [4,16]. To address these limitations and advance clinical translation, we propose the following: First, a comprehensive investigation of sexual dimorphism integrating ovariectomized models with clinical cohorts of surgical menopause patients. Second, deep functional phenotyping of NTRK1/IGF2 haplotypes in existing pain GWAS resources. Third, longitudinal CSF biomarker profiling measuring dynamic IGF2 trajectories in high-risk surgical cohorts.

## 5. Conclusions

This study establishes the spinal NTRK1-IGF2-ER stress signaling axis as a fundamental pathogenic driver of CPSP. We demonstrate a unidirectional regulatory hierarchy wherein NTRK1 transcriptionally governs IGF2 to amplify proteotoxic stress, a mechanism previously documented in glioblastoma but newly implicated in pain neuroscience. This maladaptive feedforward loop positions NTRK1 inhibitors and IGF2-targeted therapeutics as compelling intervention strategies. Therapeutically, our findings foreground IGF2 as a high-value yet context-dependent target: while its suppression alleviates CPSP, its dualistic roles in neuroprotection demand precision approaches such as neuron-restricted delivery or epigenetic modulation to mitigate off-target risks. As ER stress emerges as a cross-disease mechanism, strategies targeting this pathway may yield broad benefits beyond pain management. This study establishes a strong foundation for comprehending the pathogenesis of CPSP and devising effective therapeutic strategies, offering valuable insights for future investigations.

## Figures and Tables

**Figure 1 biomedicines-13-01632-f001:**
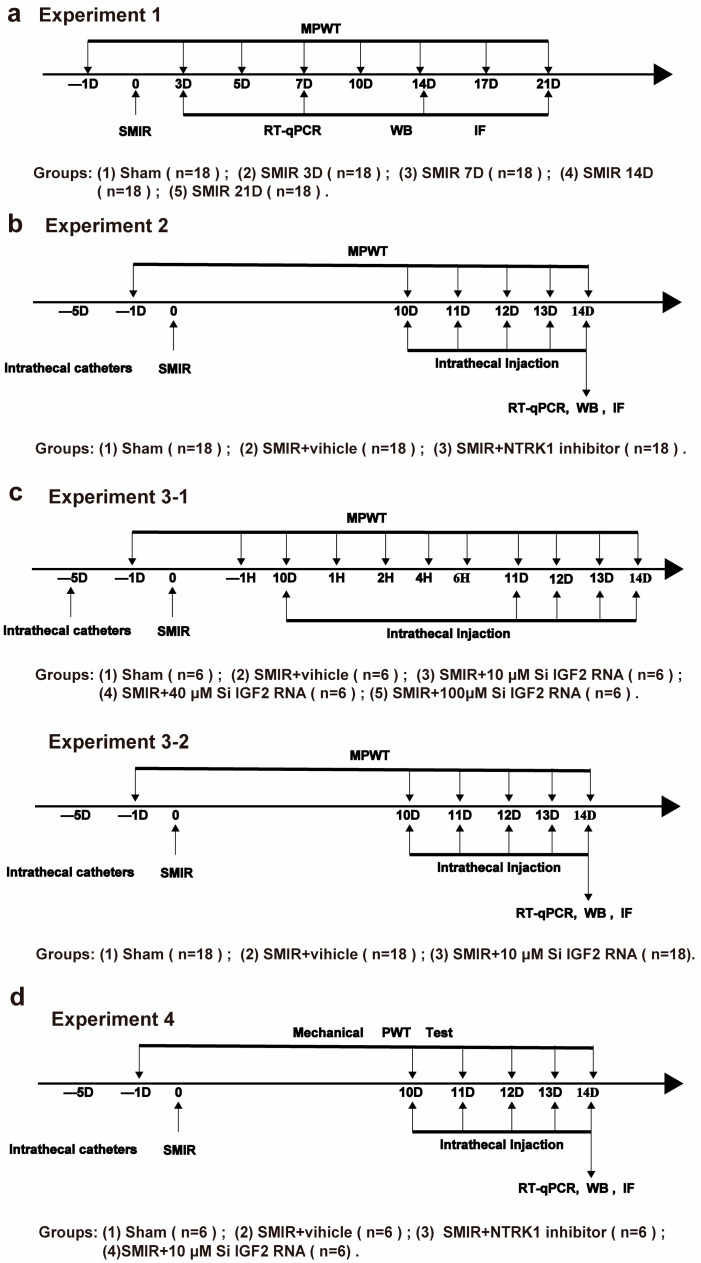
Experimental designs and animal groups. (**a**) Experiment 1: Assessment of NTRK1, ER stress, and IGF2 alterations in the spinal cord of rats with SMIR. (**b**) Experiment 2: Evaluation of NTRK1 inhibitor effects on mechanical allodynia and ER stress expression in rats with SMIR. (**c**) Experiment 3-1: Determination of the effective dose of IGF2 siRNA on mechanical allodynia in rats with SMIR; Experiment 3-2: Effect of IGF2 siRNA on mechanical allodynia and ER stress expression in SMIR rats. (**d**) Experiment 4: Comparison of the effects of NTRK1 inhibitor and IGF2 siRNA on the expression of IGF2 and NTRK1, and their relationship in rats with SMIR. SMIR: skin–muscle incision and retraction; MPWT: mechanical paw withdrawal threshold measurement; RT-qPCR: real-time quantitative polymerase chain reaction; WB: Western blot; IF: Immunofluorescence staining.

**Figure 2 biomedicines-13-01632-f002:**
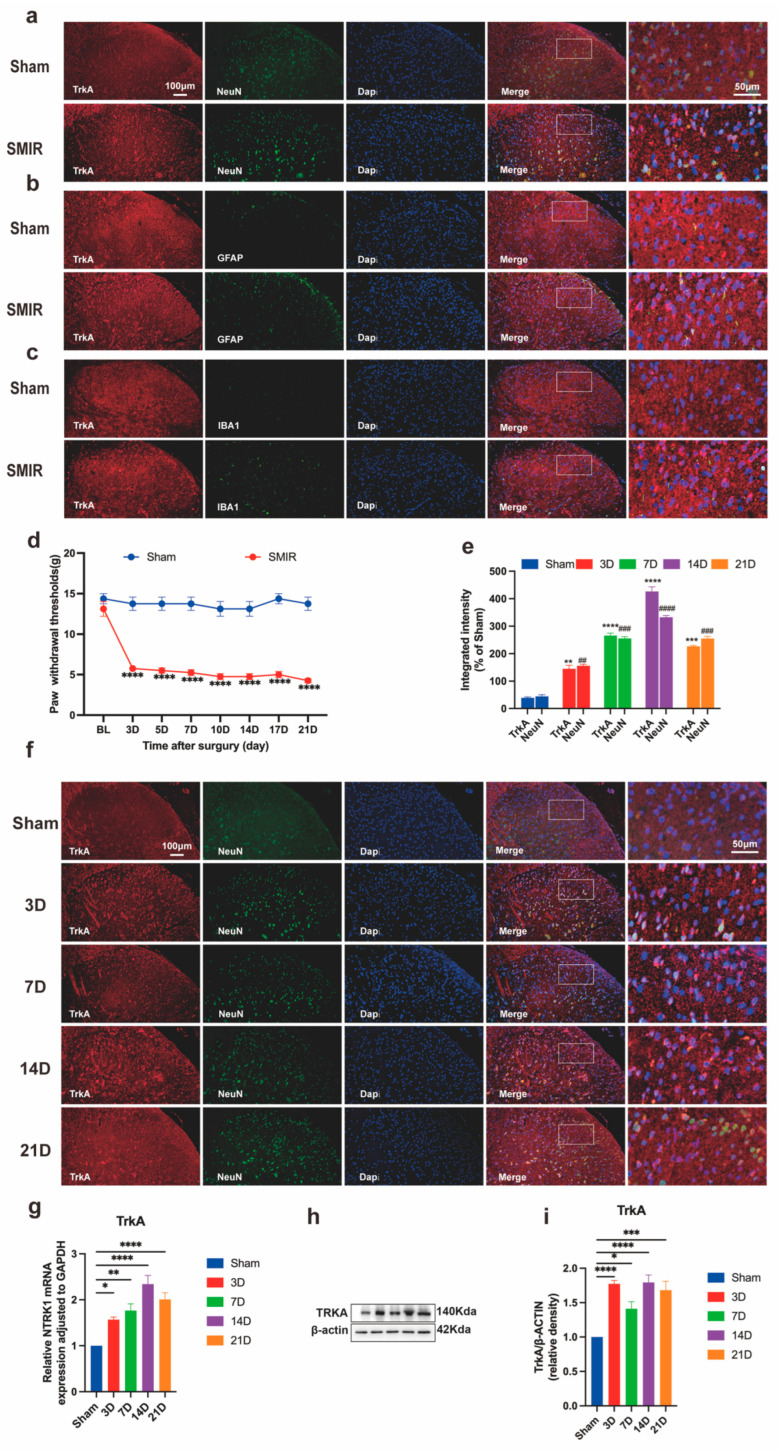
Temporal upregulation of NTRK1 in spinal dorsal horn neurons following SMIR surgery. (**a**–**c**) Representative immunofluorescence images showing NTRK1 (TrkA) co-localization with (**a**) neuronal (NeuN), (**b**) astrocytic (GFAP), and (**c**) microglial (IBA1) markers in spinal cord sections. Scale bars: 100 μm (overview), 50 μm (insets). (**d**) Mechanical hypersensitivity progression measured by paw withdrawal threshold (PWT) **** *p* < 0.0001 compared with the Sham group, n = 6/group). (**e**) Quantification of TrkA/NeuN co-localization intensity across postoperative timepoints (** *p* < 0.01, *** *p* < 0.001, **** *p* < 0.0001 compared with the Sham group; ^##^ *p* < 0.01, ^###^ *p* < 0.001, ^####^ *p* < 0.0001 compared with the Sham group; *n* = 3/group) (**f**) Immunofluorescence visualization of NTRK1 expression dynamics in dorsal horn. Scale bars: 100 μm (overview), 50 μm (insets). (**g**) qPCR analysis of *NTRK1* transcriptional regulation (* *p* < 0.05, ** *p* < 0.01, **** *p* < 0.0001 compared with the Sham group, *n* = 6/group).(**h**,**i**) Western blot assessment of NTRK1 protein expression (representative blots and quantification, * *p* < 0.05, *** *p* < 0.001, **** *p* < 0.0001 compared with the Sham group, n = 6/group). SMIR, Skin/Muscle Incision and Retraction.

**Figure 3 biomedicines-13-01632-f003:**
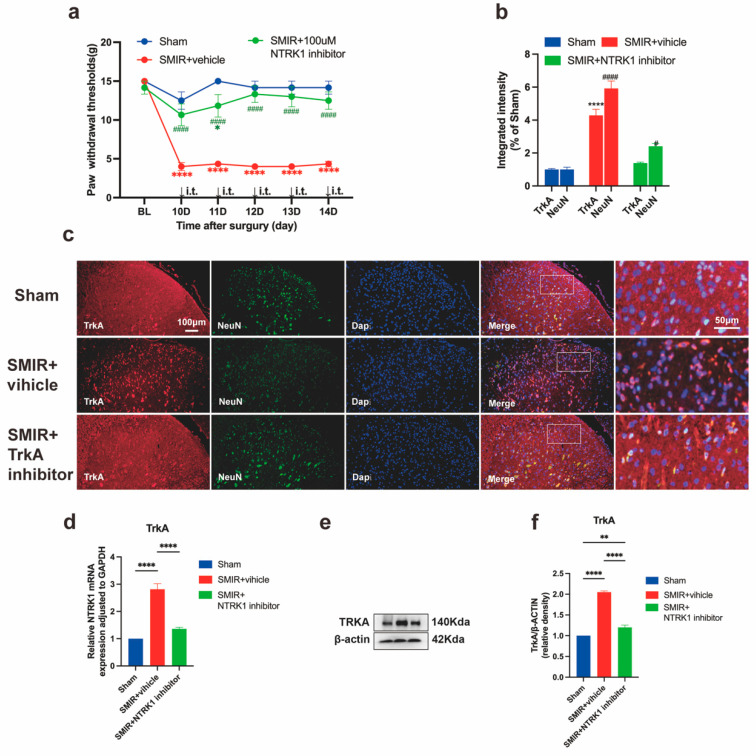
Therapeutic impact of intrathecal NTRK1 inhibition on mechanical allodynia and NTRK1 expression in SMIR rats. (**a**) Mechanical hypersensitivity measured by paw withdrawal threshold (PWT) following intrathecal administration of NTRK1 inhibitor (green) versus vehicle (red) (* *p* < 0.05, **** *p* < 0.0001 compared with the Sham group; ^####^ *p* < 0.001 compared with the SMIR + Vehicle group; n = 6/group). (**b**) Quantification of TrkA/NeuN co-localization intensity across treatment groups (**** *p* < 0.0001 compared with the Sham group; ^#^ *p* < 0.05, ^####^ *p* < 0.0001 compared with the Sham group; *n* = 3/group). (**c**) Representative immunofluorescence images showing NTRK1 expression in the spinal dorsal horn. Scale bars: 100 μm (overview), 50 μm (insets). (**d**) qPCR analysis of *NTRK1* transcriptional regulation (**** *p* < 0.0001; *n* = 6/group). (**e**,**f**) Western blot assessment of NTRK1 protein expression (representative blots and quantification, ** *p* < 0.01, **** *p* < 0.0001; *n* = 6/group). SMIR, Skin/Muscle Incision and Retraction.

**Figure 4 biomedicines-13-01632-f004:**
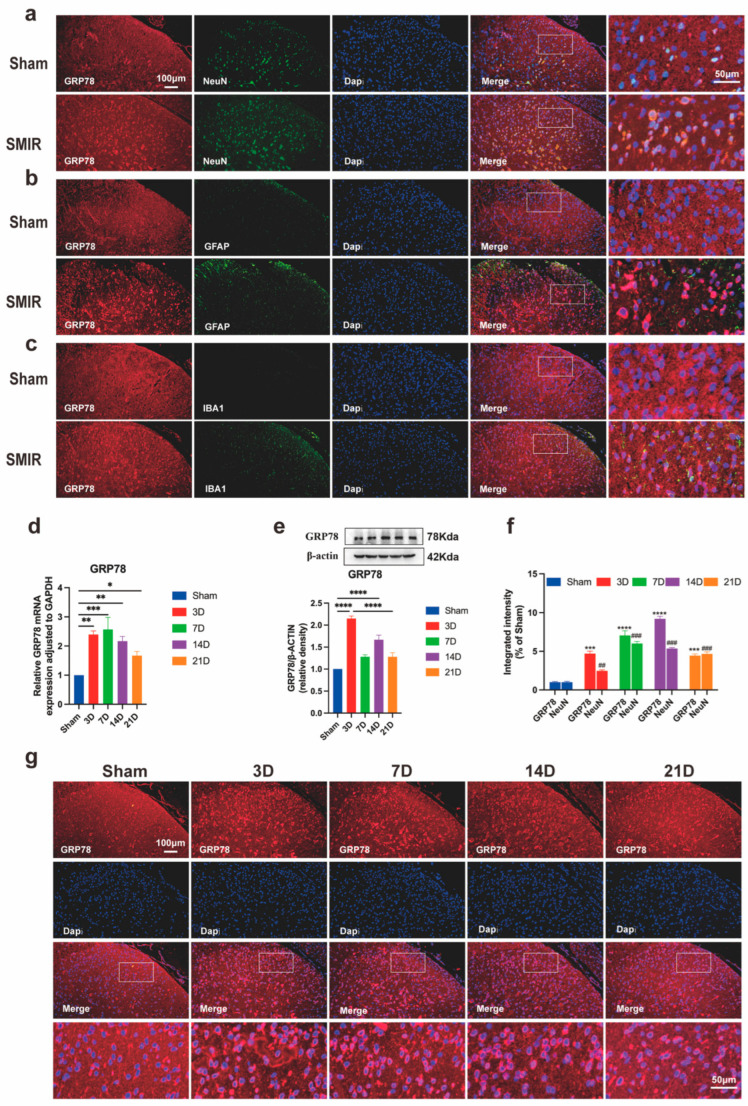
Temporal upregulation of GRP78 in spinal dorsal horn following SMIR surgery. (**a**–**c**) Representative immunofluorescence images showing GRP78 co-localization with (**a**) neuronal (NeuN), (**b**) astrocytic (GFAP), and (**c**) microglial (IBA1) markers in spinal cord sections. Scale bars: 100 μm (overview), 50 μm (insets). (**d**) qPCR analysis of *GRP78* transcriptional regulation (* *p* < 0.05, ** *p* < 0.01, *** *p* < 0.001; n = 6/group). (**e**) Western blot assessment of GRP78 protein expression (representative blots and quantification, **** *p* < 0.0001; n = 6/group). (**f**) Quantification of GRP78/NeuN co-localization intensity across postoperative timepoints (*** *p* < 0.001, **** *p* < 0.0001 compared with the Sham group; ^##^ *p* < 0.01, ^###^ *p* < 0.001, compared with the Sham group; n = 3/group). (**g**) Immunofluorescence visualization of GRP78 expression dynamics in the dorsal horn. Scale bars: 100 μm (overview), 50 μm (insets). SMIR, Skin/Muscle Incision and Retraction.

**Figure 5 biomedicines-13-01632-f005:**
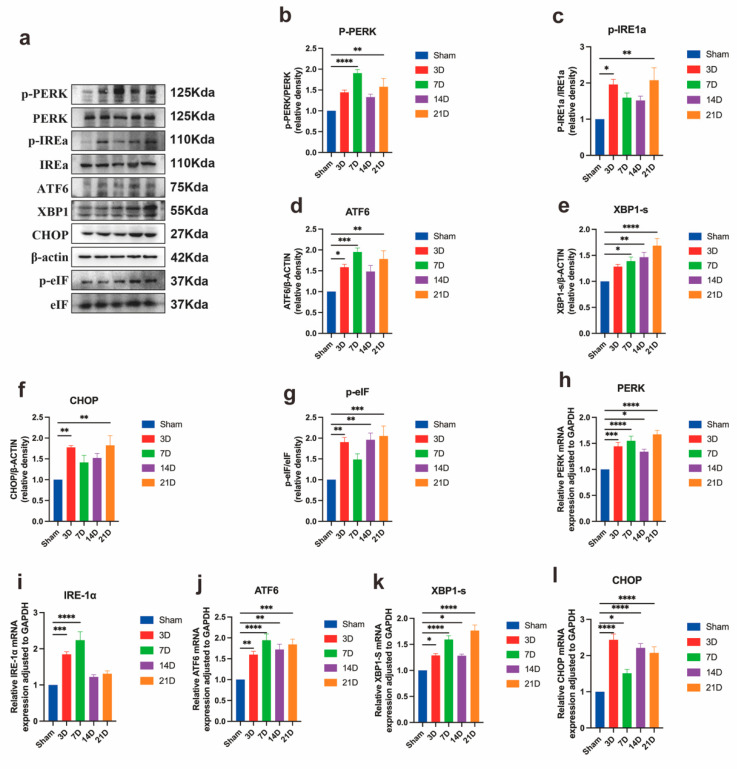
Temporal activation of endoplasmic reticulum stress pathways in spinal cord of SMIR rats. (**a**) Representative western blots of UPR signaling proteins. (**b**–**g**) Quantification of key ER stress markers: p-PERK, p-IRE1α, p-eIF2α, XBP1s, ATF6, and CHOP. Temporal profiles show progressive upregulation from days 3 to 21 post-SMIR (* *p* < 0.05, ** *p* < 0.01, *** *p* < 0.001, **** *p* < 0.0001; n = 6/group). (**h**–**l**) qPCR analysis of corresponding transcriptional regulation: *PERK*, *IRE1α*, *XBP1s*, *ATF6*, and *CHOP* (* *p* < 0.05, ** *p* < 0.01, *** *p* < 0.001, **** *p* < 0.0001; n = 6/group). SMIR, Skin/Muscle Incision and Retraction.

**Figure 6 biomedicines-13-01632-f006:**
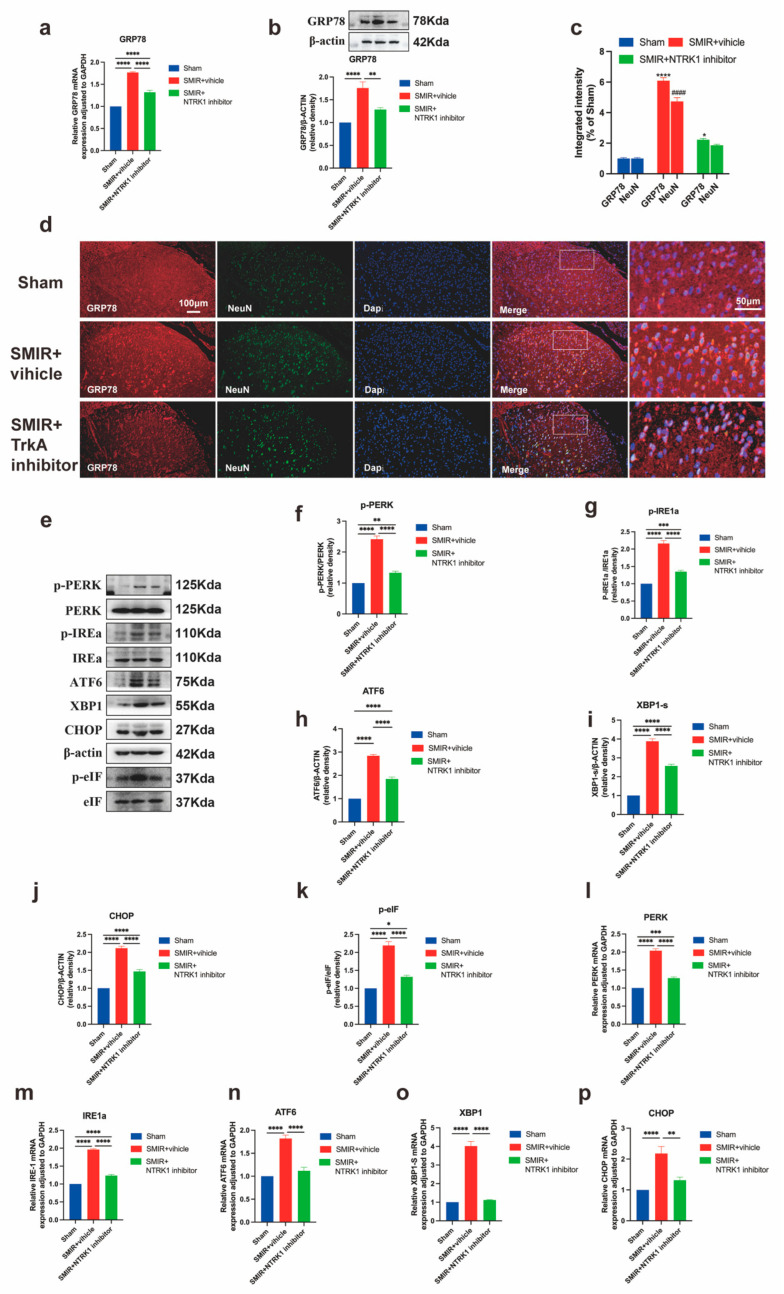
NTRK1 inhibition suppresses ER stress activation in SMIR rats. (**a**) qPCR analysis of *GRP78* transcriptional regulation following intrathecal NTRK1 inhibitor administration (**** *p* < 0.0001; *n* = 6/group). (**b**) Western blot assessment of GRP78 protein expression (representative blots and quantification, ** *p* < 0.01, **** *p* < 0.0001; *n* = 6/group). (**c**) Quantification of GRP78/NeuN co-localization intensity across treatment groups (* *p* < 0.05, **** *p* < 0.0001 compared with the Sham group; ^####^
*p* < 0.0001 compared with the Sham group; *n* = 3/group). (**d**) Immunofluorescence visualization of GRP78 expression in the spinal dorsal horn. Scale bars: 100 μm (overview), 50 μm (insets). (**e**) Representative western blots of UPR signaling proteins. (**f**–**k**) Quantification of key ER stress markers: p-PERK, p-IRE1α, p-eIF2α, XBP1s, ATF6, and CHOP. (* *p* < 0.05, ** *p* < 0.01, *** *p* < 0.001, **** *p* < 0.0001; *n* = 6/group). (**l**–**p**) qPCR analysis of corresponding transcriptional changes: *PERK*, *IRE1α*, *XBP1s*, *ATF6*, and *CHOP* (** *p* < 0.01, *** *p* < 0.001, **** *p* < 0.0001; *n* = 6/group).

**Figure 7 biomedicines-13-01632-f007:**
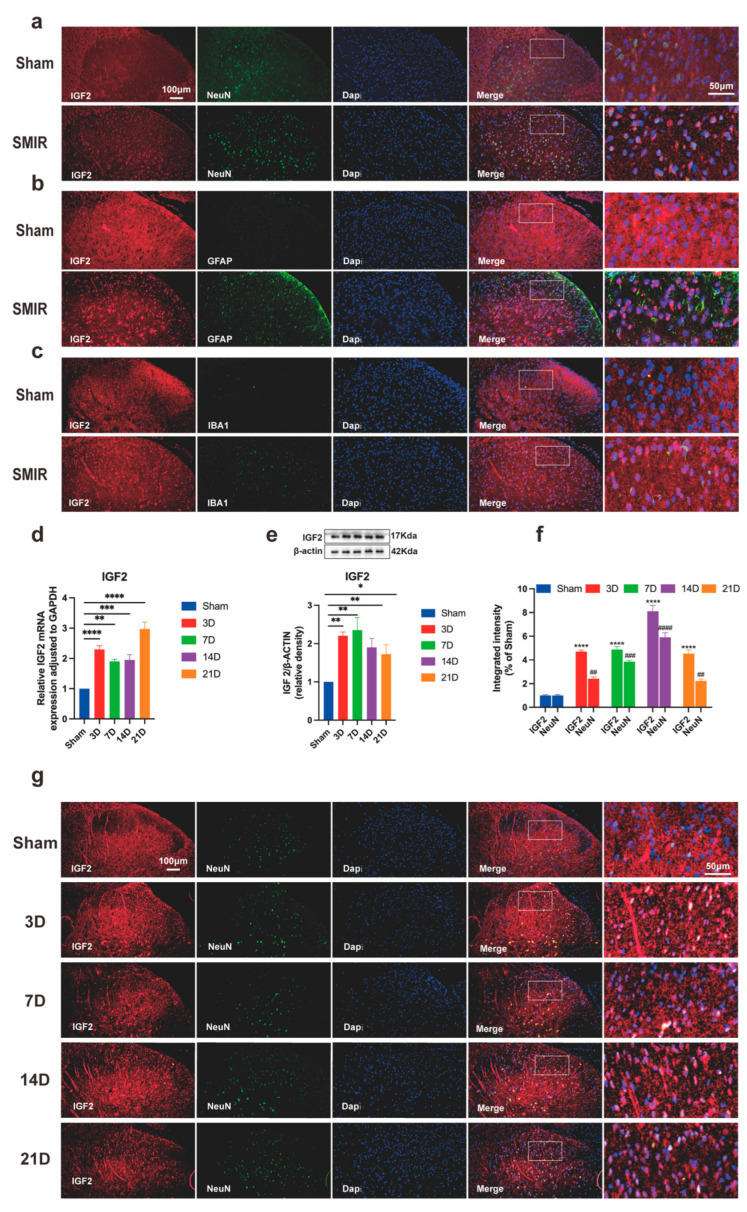
Temporal upregulation of IGF2 in spinal dorsal horn following SMIR surgery. (**a**–**c**) Representative immunofluorescence images showing IGF2 co-localization with (**a**) neuronal (NeuN), (**b**) astrocytic (GFAP), and (**c**) microglial (IBA1) markers in spinal cord sections. Scale bars: 100 μm (overview), 50 μm (insets). (n = 3/group). (**d**) qPCR analysis of *IGF2* transcriptional regulation (** *p* < 0.01, *** *p* < 0.001, **** *p* < 0.0001; n = 6/group). (**e**) Western blot assessment of IGF2 protein expression (representative blots and quantification, * *p* < 0.05 ** *p* < 0.01; n = 6/group). (**f**) Quantification of IGF2/NeuN co-localization intensity across postoperative timepoints (**** *p* < 0.0001 compared with the Sham group; ^##^ *p* < 0.01, ^###^ *p* < 0.001, ^####^ *p* < 0.0001 compared with the Sham group; n = 3/group). (**g**) Immunofluorescence visualization of IGF2 expression dynamics in the dorsal horn. Scale bars: 100 μm (overview), 50 μm (insets). (n = 3/group).

**Figure 8 biomedicines-13-01632-f008:**
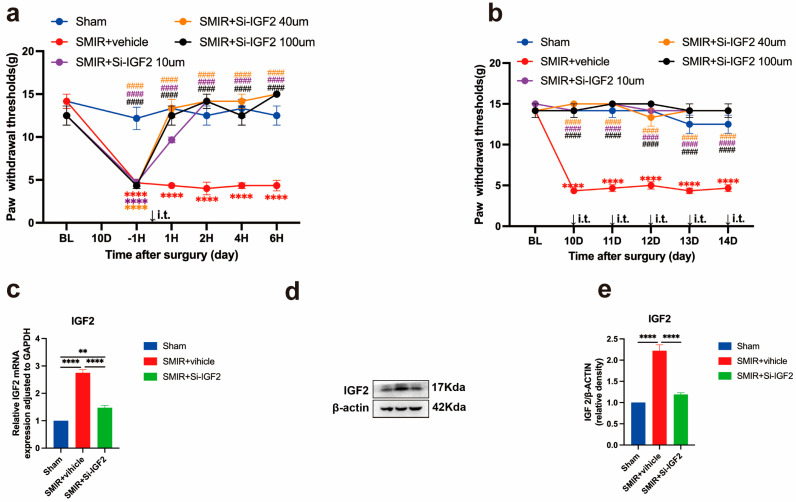
Therapeutic impact of IGF2 silencing on mechanical allodynia and IGF2 expression in SMIR rats. (**a**) Dose-dependent effects of single intrathecal IGF2 siRNA administration (10 μM, 40 μM, 100 μM) on mechanical hypersensitivity. (**** *p* < 0.0001 compared with the Sham group; ^####^ *p* < 0.0001 compared with the SMIR + Vehicle group; n = 6/group). (**b**) Dose-response profile of continuous intrathecal IGF2 siRNA treatment on pain behavior. (**** *p* < 0.0001 compared with the Sham group; ^####^ *p* < 0.0001 compared with the SMIR + Vehicle group; n = 6/group). (**c**) qPCR analysis of IGF2 transcriptional suppression. (** *p* < 0.01, **** *p* < 0.0001 compared with the Sham group; n = 6/group). (**d**,**e**) Western blot assessment of IGF2 protein expression (representative blots and quantification). (**** *p* < 0.0001 compared with the Sham group; n = 6/group).

**Figure 9 biomedicines-13-01632-f009:**
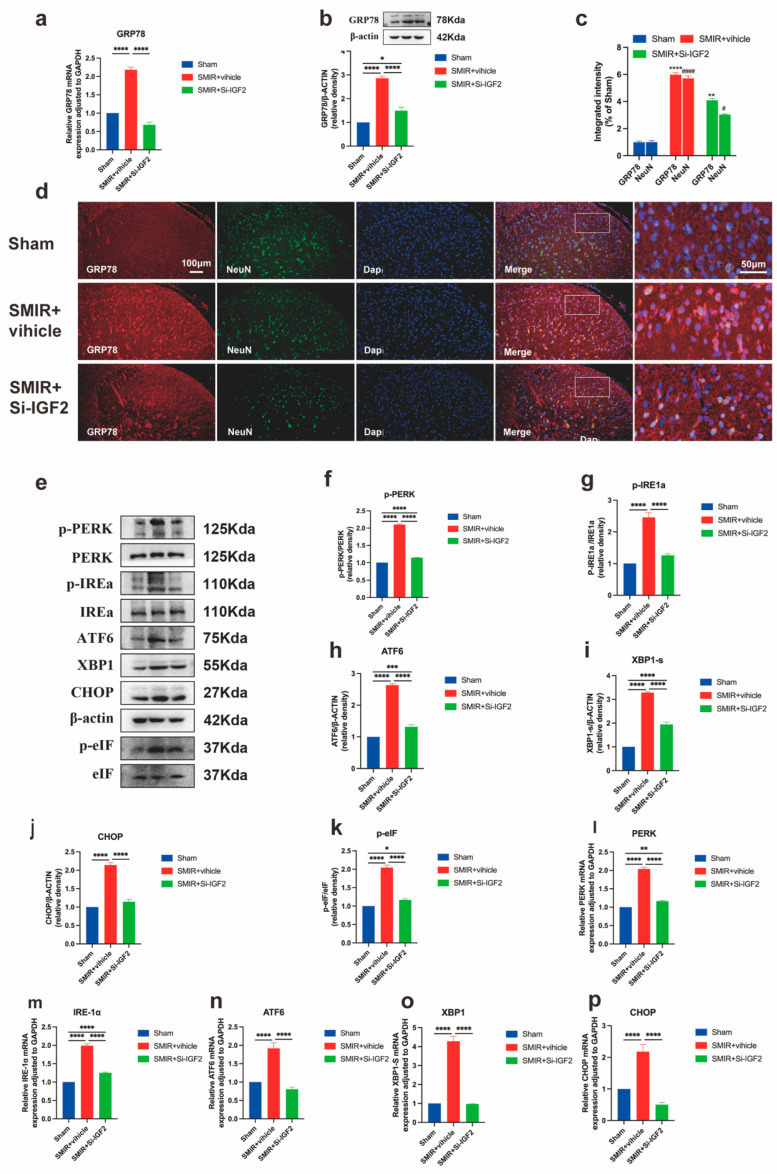
IGF2 silencing attenuates ER stress activation in SMIR rats. (**a**) qPCR analysis of *GRP78* transcriptional suppression following IGF2 siRNA administration (**** *p* < 0.0001 compared with the sham group; n = 6/group). (**b**) Western blot assessment of GRP78 protein expression (representative blots and quantification, * *p* < 0.05, **** *p* < 0.0001 compared with the sham group; n = 6/group). (**c**) Quantification of GRP78/NeuN co-localization intensity across treatment groups (** *p* < 0.01, **** *p* < 0.0001 compared with the Sham group; ^#^ *p* < 0.05, ^####^ *p* < 0.0001 compared with the Sham group; n = 3/group). (**d**) Immunofluorescence visualization of GRP78 expression in the spinal dorsal horn. Scale bars: 100 μm (overview), 50 μm (insets). (**e**) Representative western blots of UPR signaling proteins. (**f**–**k**) Quantification of key ER stress markers: p-PERK, p-IRE1α, p-eIF2α, XBP1s, ATF6, and CHOP (* *p* < 0.05, *** *p* < 0.001, **** *p* < 0.0001; n = 6/group). (**l**–**p**) qPCR analysis of corresponding transcriptional changes: *PERK*, *IRE1α*, *XBP1s*, *ATF6*, and *CHOP* (** *p* < 0.01, **** *p* < 0.0001; n = 6/group).

**Figure 10 biomedicines-13-01632-f010:**
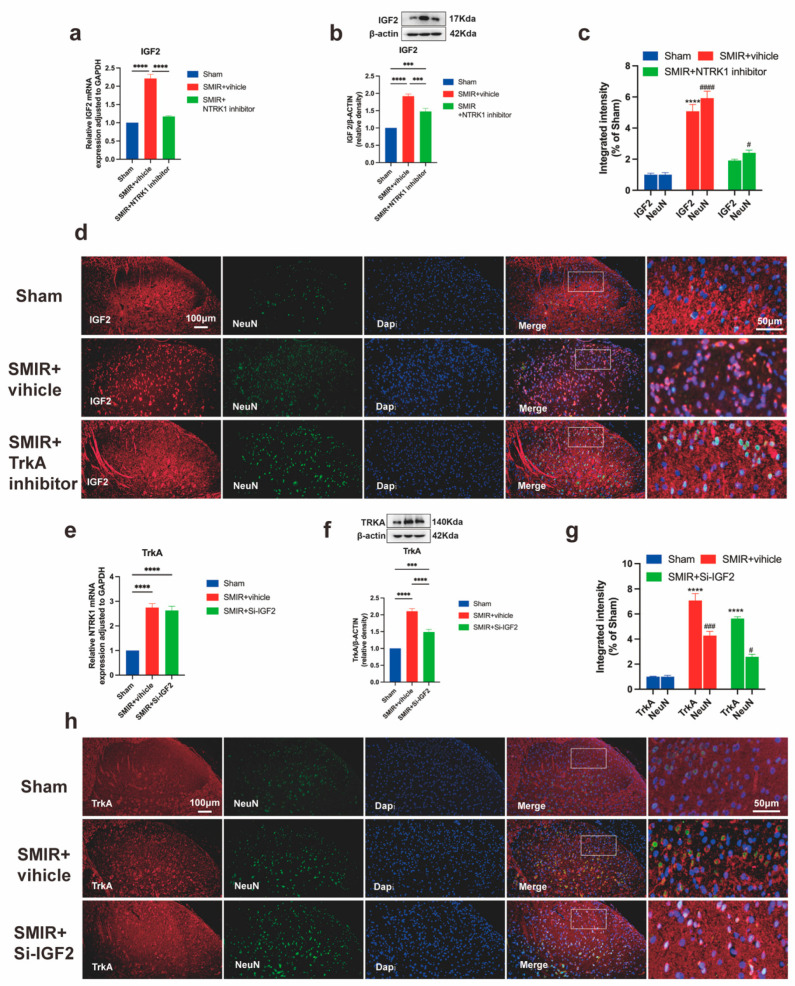
Unidirectional regulation within the NTRK1-IGF2 axis in CPSP. (**a**) qPCR analysis of *IGF2* transcriptional changes following NTRK1 inhibition (**** *p* < 0.0001; n = 6/group). (**b**) Western blot assessment of IGF2 protein expression after NTRK1 inhibitor treatment (representative blots and quantification, *** *p* < 0.001, **** *p* < 0.0001; n = 6/group). (**c**) Quantification of IGF2/NeuN co-localization intensity across treatment groups (**** *p* < 0.0001 compared with the Sham group; ^#^ *p* < 0.05, ^####^ *p* < 0.0001 compared with the Sham group; n = 3/group). (**d**) Immunofluorescence visualization of neuronal IGF2 expression. Scale bars: 100 μm (overview), 50 μm (insets). (**e**) qPCR analysis of *NTRK1* transcriptional changes following IGF2 silencing (**** *p* < 0.0001; n = 6/group). (**f**) Western blot assessment of TrkA protein expression after IGF2 siRNA administration (representative blots and quantification, *** *p* < 0.001, **** *p* < 0.0001; n = 6/group). (**g**) Quantification of TrkA/NeuN co-localization intensity (**** *p* < 0.0001 compared with the Sham group; ^#^ *p* < 0.05, ^###^ *p* < 0.001 compared with the Sham group; n = 3/group). (**h**) Immunofluorescence visualization of neuronal TrkA expression. Scale bars: 100 μm (overview), 50 μm (insets).

**Table 1 biomedicines-13-01632-t001:** Pharmacological interventions.

Agent	Formulation	Administration	Dose Rationale
GW441756 (NTRK1i; Selleck #S2789)	100 μM in saline + 1% DMSO	i.t. daily, days 10–14	Dose-response pilot [28]
IGF2 siRNA (Tsingke Biotechnology)	10–100 μM in 5% glucose + PEI (1:5 *w*/*w*)	i.t. single dose (day 10)	Prior efficacy in neuropathic pain [51]
IGF2 siRNA (Tsingke Biotechnology)	10 μM in 5% glucose + PEI (1:5 *w*/*w*)	i.t. daily, days 10–14	Prior efficacy in neuropathic pain [51]
Vehicle controls	Saline + 1% DMSO or PEI/glucose	Matched volumes/timing	N/A

**Table 2 biomedicines-13-01632-t002:** The primer sequences used for quantitative PCR assay.

Gene	Forward Primer Sequence (5′ to 3′)	Reverse Primer Sequence (5′ to 3′)
*GAPDH*	GAAGGTCGGTGTGAACGGAT	CCCATTTGATGTTAGCGGGAT
*NTRK1*	AGGAGGATTTGTGTGGTGTGTAT	GAGTCATTGGGCATCTGGATCTT
*IGF2*	GGGAAGTCGATGTTGGTGCT	AAGCAGCACTCTTCCACGAT
*GRP78*	GGTTGGCGGATCTACTCGAATTC	AAGAGGACACACATCAAGCAGAA
*XBP1*	CCCAGAACATCTTCCCATGGATT	CAGAGAAAGGGAGGCTGGTAAG
*CHOP*	TTCATACACCACCACACCTGAAA	TAGGGATGCAGGGTCAAGAGTAG
*ATF6*	AGCAAGATTCCAGGAGAGTGAAA	TGACATGGAGGTGGAGGGATATA
*PERK*	TTGGAAGGTCATGGCGTTTAGTA	TGGCCTCTGTACATCCCTAAGTA
*IRE1α*	TCAAGGCGATGATCTCAGACTTT	GTTGCCCTCAGAGATGACATAGT

**Table 3 biomedicines-13-01632-t003:** The detailed information on antibodies.

Antibodies	Source	Identifier	Dilution	Applications
Anti-TrkA antibody	ABclonal	A15618	1:1000,1:50	WB,IF
Anti-IGF2 antibody	CST	Ab9574	1:2000	WB
Anti-IGF2 antibody	ABclonal	A2086	1:50	IF
Anti-GRP78 antibody	Affinity	Cat# AF5366	1:1000,1:50	WB,IF
Anti-β-actin antibody	Proteintech	Cat# 81115-1-RR	1:5000	WB
Anti-p-PERK antibody	Affinity	Cat# DF7576	1:100	WB
Anti-PERK antibody	ABclonal	Cat# A21255	1:1000	WB
Anti-p-IRE1α antibody	Affinity	Cat# AF7150	1:100	WB
Anti-IRE1α antibody	Proteintech	Cat# 27528-1-AP	1:1000	WB
Anti-p-eIF antibody	Affinity	Cat# AF3087	1:100	WB
Anti-eIF antibody	Proteintech	Cat# 11170-1-AP	1:1000	WB
Anti-XBP1 antibody	Proteintech	Cat# 24868-1-AP	1:1000	WB
Anti-CHOP antibody	ABclonal	A20987	1:1000	WB
Anti-ATF6 antibody	Affinity	Cat# DF6009	1:1000	WB
Anti-NeuN antibody	Abcam	Ab104224	1:200	IF
Anti-GFAP antibody	CST	Cat# 3670	1:200	IF
Anti-IBA1 antibody	Abcam	Ab5076	1:200	IF
oraLite594 anti-rabbit IgG	Proteintech	Cat# SA00013-8	1:200	IF
oraLite488 anti-mouse IgG	Proteintech	Cat# SA00013-5	1:200	IF
FITC affinipure anti-goat IgG	Proteintech	Cat# SA00003-3	1:200	IF
Anti-rabbit IgG HRP	Abbkine	Cat# A21020	1:5000	WB
Anti-goat IgG HRP	ABclonal	Cat# AS031	1:5000	WB

## Data Availability

The original contributions presented in this study are included in the article. Further data are available from the corresponding author upon reasonable request at ourpain@163.com.

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
