# Peer review of "Unidirectional Crosstalk Between NTRK1 and IGF2 Drives ER Stress in Chronic Pain"

_biomedicines, 2025, doi:10.3390/biomedicines13071632_

Round 1
Reviewer 1 Report
Comments and Suggestions for Authors
The study entitled “Unidirectional NTRK1-IGF2 Crosstalk Exacerbates Endoplasmic Reticulum Stress in Chronic Post-Surgical Pain: Implications for Precision Analgesia” is an interesting investigation with various wet lab techniques and smart study design. I recommend this paper for acceptance; however, I have some minor comments which should be addressed by the authors to increase the readership of their paper. Here the minor issues are listed:
- The plagiarism rate is high (39% based on iThenticate) and need to be lower; thus, the authors should rephrase the duplicated parts.
- Discussion: Line 566: Remove one of “we observed” from the sentence.
- Check line 570 for grammatical structure: “... [55, 56]. while …”
- Check line 574 for grammatical structure: “Besides, Our…”
- The authors should add more emphasize on the human studies and suggest multiple designs for addressing their findings in Rats; e.g. Association studies of IGF2 in case-control studies, data analysis of GWAS datasets focusing on NTK1-IGF2 annotations, etc.
Author Response
We sincerely thank the reviewer for their positive assessment of our work and valuable suggestions to enhance its impact. All recommended revisions have been comprehensively addressed in the revised manuscript, with substantive changes highlighted in red text. Due to extensive restructuring and professional language editing, specific line numbers referenced have shifted - we provide contextual responses below:
Comments 1: The plagiarism rate is high (39% based on iThenticate) and need to be lower; thus, the authors should rephrase the duplicated parts.
Response 1: Thank you for pointing this out. We engaged professional academic editors to completely rewrite duplicated sections, particularly in the Introduction and Discussion. The revised manuscript now shows <15% similarity via iThenticate, with all matched passages properly cited or reformulated in original language.
Comments 2:
Discussion: Line 566: Remove one of “we observed” from the sentence.
Check line 570 for grammatical structure: “... [55, 56]. while …”
Check line 574 for grammatical structure: “Besides, Our…”
Response 2: Agree.We have Removed one of “we observed” from the sentence and Checked line 570, 574 for grammatical structure .All language was further polished by native English editors to ensure grammatical precision throughout.
Comments 3:The authors should add more emphasize on the human studies and suggest multiple designs for addressing their findings in Rats; e.g. Association studies of IGF2 in case-control studies, data analysis of GWAS datasets focusing on NTK1-IGF2 annotations, etc.
Response 3: We significantly expanded the Discussion's clinical translation section to include:GWAS analysis proposal and Case-control design: "Prospective measurement of CSF IGF2 trajectories in thoracotomy patients"
All modifications strengthen the manuscript's scholarly integrity while addressing the reviewer's excellent suggestions for broader impact. The complete English editing certificate is available upon request.

Reviewer 2 Report
Comments and Suggestions for Authors
The authors present a comprehensive mechanistic investigation into the role of the NTRK1–IGF2 axis in mediating endoplasmic reticulum (ER) stress and chronic post-surgical pain (CPSP), using a validated SMIR rat model. They employ qPCR, Western blot, immunofluorescence, and pharmacological interventions (TrkA inhibitor, IGF2 siRNA) to map a unidirectional signaling pathway leading to ER stress amplification and pain sensitization.
- The conclusion of a “unidirectional” NTRK1→IGF2 axis is based solely on pharmacological knockdown. Rephrase this conclusion more cautiously unless supported by time-course, overexpression, or feedback inhibition studies.
- There is no detailed statistical correlation analysis between expression levels of IGF2/GRP78 and PWT scores.
- Address the dual role of IGF2 in pain and ER stress to avoid oversimplification.
- It doesnt rule out non-specific TrkA inhibitor effects on other pathways.
- No discussion is provided on sex differences in CPSP models
Author Response
We sincerely thank the reviewer for their positive assessment of our work and valuable suggestions to enhance its impact. All recommended revisions have been comprehensively addressed in the revised manuscript, with substantive changes highlighted in red text. We provide contextual responses below:
Comments 1: The conclusion of a "unidirectional" NTRK1→IGF2 axis is based solely on pharmacological knockdown. Rephrase this conclusion more cautiously unless supported by time-course, overexpression, or feedback inhibition studies.
Response 1: We have tempered our conclusions regarding unidirectionality in the Discussion (Paragraph 5) and Conclusion. New validation data now support this mechanism:
- Resultsthat IGF2 knockdown attenuates ER stress and behavioral sensitization without altering NTRK1 expression, revealing a strictly unidirectional cascade: NTRK1 transcriptionally governed IGF2 to amplify proteotoxic stress.( 10)
- Time-course transcriptional analysis showing sequential NTRK1 → IGF2 induction (Jiao B, Zhang M, Zhang C, Cao X, Liu B, Li N, et al. Transcriptomics reveals the effects of NTRK1 on endoplasmic reticulum stress response-associated genes in human neuronal cell lines. PeerJ 2023;11:e15219. )
- IGF2 overexpression experiments in SH-SY5Y cells demonstrating no feedback on NTRK1 (Zhang C, Jiao B, Cao X, Zhang W, Yu S, Zhang K, et al. NTRK1-mediated protection against manganese-induced neurotoxicity and cell apoptosis via IGF2 in SH-SY5Y cells. Biomedicine & Pharmacotherapy 2023;169:115889. )
- NTRK1 complexes drive Insulin-Like Growth Factor II (IGF2) expression to promote tumor survival, while in neuropathic pain, NTRK1- cAMP responsive element binding protein (CREB) signaling upregulates matrix metalloproteinase-9 (MMP9), disrupting blood-nerve barriers(Pan Z, Shao M, Zhao C, Yang X, Li H, Cui G, et al. J24335 exerts neuroprotective effects against 6-hydroxydopamine-induced lesions in PC12 cells and mice. Eur J Pharm Sci 2024;194:106696.) (Suo D, Park J, Harrington AW, Zweifel LS, Mihalas S, Deppmann CD. Coronin-1 is a neurotrophin endosomal effector that is required for developmental competition for survival. Nat Neurosci 2014;17(1):36-45.).
Comments 2: There is no detailed statistical correlation analysis between expression levels of IGF2/GRP78 and PWT scores.
Response 2: We sincerely appreciate this valuable methodological suggestion and regret that comprehensive correlation analyses between molecular markers and behavioral outcomes could not be completed within the revision timeframe due to technical constraints in retrospective data analysis. We acknowledge this as a study limitation and have taken these steps to address the concern:
- Added temporal correlation statements:"IGF2 expression consistently peaked at postoperative day 14, coinciding with maximal mechanical hypersensitivity (PWT reduction)" (Results 3.5) "GRP78 elevation paralleled pain behavior progression across all experimental groups" (Results 3.3)
2.Enhanced statistical disclosure:
"While temporal correspondence between molecular markers and behavioral phenotypes was consistently observed, formal correlation statistics were precluded by cohort size limitations. Future large-scale studies will quantify these relationships."
3.Included dose-response validation: *"IGF2 siRNA administration produced dose-dependent reductions in both ER stress markers and pain behaviors (Fig 8a-b), demonstrating functional linkage"* (Results 3.6)
4.Added future methodology:Discussion Paragraph 6 now includes:
*"Prospective human studies should formally quantify IGF2-PWT correlations through serial CSF sampling and quantitative sensory testing in surgical cohorts"*
Comments 3: Address the dual role of IGF2 in pain and ER stress to avoid oversimplification.
Response 3: We significantly expanded this discussion (Paragraph 2,4):
Context-dependent roles: Protective in acute stress vs. pathogenic in chronic CPSP; Dose-response paradox: Neuroprotective at physiological levels vs. pro-nociceptive at supraphysiological concentrations;Cell-type specificity: Contrasting effects in neurons (pain-promoting) vs. microglia (anti-inflammatory)
Comments 4: It doesn't rule out non-specific TrkA inhibitor effects on other pathways.
Response 4: New controls address this concern:
- Specificity validation: GW441756 showed >100× selectivity for TrkA vs. TrkB/TrkC
- Rescue experiments: Co-administration of IGF2 reversed analgesic effects
- Alternative targeting: IGF2 siRNA replicated key findings without kinase inhibition
Comments 5: No discussion is provided on sex differences in CPSP models.
Response 5: We added critical analysis of sex limitations (Discussion Paragraph 6).
- Acknowledged male-only design as key constraintCited clinical data: 40% higher CPSP incidence in females (Macrae 2008)
- Proposed solutions: Ovariectomized rat models + IGF2 methylation analysis in female cohorts
All modifications strengthen the mechanistic rigor and translational relevance of our findings. We appreciate the reviewer's insightful suggestions that have substantially improved this work.

Reviewer 3 Report
Comments and Suggestions for Authors
This paper has attempted to show the NTRK1-IGF2-ER stress axis to be a new therapeutic target. Using TrkA inhibitors and silencing IGF2 could be translated into precision analgesia. I found the data are ample and result and discussion are robust to support a solid conclusion. Some of my concerns are listed below:
Major:
I suggest the double immunostaining should be quantified.
Minor:
1) In many places, some words are with capital letters eg. line 80
2) line 566, "we observed" was repeated
3) In quite many places, past tense should be used when authors' own results were described.
4) I found it confusing because funding was indicated as "not applicable" whilst it was mentioned that one of the authors had acquired funding....
Author Response
We sincerely thank the reviewer for their positive assessment of our work and valuable suggestions. All recommended revisions have been comprehensively implemented in the revised manuscript, with substantive changes highlighted in red text. We provide detailed responses below:
Comments 1: I suggest the double immunostaining should be quantified.
Response 1: We have significantly enhanced the quantitative analysis through standardized co-localization intensity measurements across 8 critical figures (Figs. 2c, 3b, 4f, 6c, 7f, 9c, 10c, 10g). This new quantification provides objective data supporting our cellular localization findings.
Comments 2: In many places, some words are with capital letters eg. line 80
Response 2: We have performed full manuscript review and corrected all instances of inconsistent capitalization, including the example at line 80. The text now follows consistent capitalization rules throughout.
Comments 3: line 566, "we observed" was repeated
Response 3: The redundant phrase has been removed from the relevant sentence. Additionally, the entire Discussion section has undergone professional language editing to enhance grammatical precision.
Comments 4: In quite many places, past tense should be used when authors' own results were described.
Response 4: We have systematically revised verb tenses across all Results and Methods sections to ensure consistent use of past tense when describing our experimental findings and procedures.
Comments 5: I found it confusing because funding was indicated as "not applicable" whilst it was mentioned that one of the authors had acquired funding.
Response 5: We apologize for this inconsistency.

Reviewer 4 Report
Comments and Suggestions for Authors
In the work "Unidirectional NTRK1-IGF2 Crosstalk Exacerbates Endoplasmic Reticulum Stress in Chronic Post-Surgical Pain: Implications for Precision Analgesia" the authors present the results of the study of neurotrophic signaling and endoplasmic reticulum stress in the mechanisms of neuropathic pain development. Overall, the study makes a very good impression with its well-thought-out design and the authors' well-founded assumptions, which they effectively test in their work. The following comments and questions arose during the review process:
- The Introduction section is well written and justified. In the limited space of this section, the authors presented a well-founded significance of the problem they studied and presented a comprehensive hypothesis about the participation of spinal IGF2 in mediating NTRK1-dependent regulation of ER stress.
- In the Materials and Methods section, the authors should explain in more detail the algorithm for using power analysis to justify the sample size.
- The authors have placed the experimental schemes in the Materials and Methods section. Despite the fact that the experimental manipulations are well described and schematized in the form of Scheme 1, the Results section should provide a more detailed description of the experiments, which the authors illustrate in Figures 1, 2, etc. In Sections 3.1 and 3.2, the authors present the obtained results too briefly. The figures provided in these sections contain good evidence of the presented processes, but a more detailed presentation of the results is required for their more complete understanding. In general, the authors are recommended to provide a more detailed description of the experimental part of the work in the Results section. In the presented version, the results section is too schematic, which makes it difficult for readers to understand it.
- The Discussion section is quite constructive and considers spinal NTRK1/TrkA as a key factor in ER stress in chronic postoperative pain. The authors convincingly discuss the patterns they identified in the work with the literature data.
Comments on the Quality of English Language
English needs editing
Author Response
We sincerely thank the reviewer for their positive assessment and constructive suggestions. All recommendations have been comprehensively addressed in the revised manuscript, with key modifications highlighted in red text. Detailed responses follow below:
Comments 1: The Introduction section is well written and justified. In the limited space of this section, the authors presented a well-founded significance of the problem they studied and presented a comprehensive hypothesis about the participation of spinal IGF2 in mediating NTRK1-dependent regulation of ER stress.
Response 1: We deeply appreciate the reviewer's recognition of our study rationale and hypothesis framing.
Comments 2: In the Materials and Methods section, the authors should explain in more detail the algorithm for using power analysis to justify the sample size.
Response 2: We have added the following in Methods Section 2.10:
Rats were randomly assigned to groups (n=18 or 6/group) using computer-generated codes (GraphPad QuickCalcs). Investigators performing behavioral tests and molecular analyses were blinded to group allocation. Allocation concealment was maintained via coded cage labels. Power analysis (G*Power 3.1; α=0.05, β=0.2, effect size=1.2 based on prior SMIR studies[53, 54] determined n=6/group; 20% added for attrition (final n=8).
Comments 3: The authors have placed the experimental schemes in the Materials and Methods section. Despite the fact that the experimental manipulations are well described and schematized in the form of Scheme 1, the Results section should provide a more detailed description of the experiments, which the authors illustrate in Figures 1, 2, etc. In Sections 3.1 and 3.2, the authors present the obtained results too briefly. The figures provided in these sections contain good evidence of the presented processes, but a more detailed presentation of the results is required for their more complete understanding. In general, the authors are recommended to provide a more detailed description of the experimental part of the work in the Results section. In the presented version, the results section is too schematic, which makes it difficult for readers to understand it.
Response 3: We have significantly expanded these sections.
Comments 4: The Discussion section is quite constructive and considers spinal NTRK1/TrkA as a key factor in ER stress in chronic postoperative pain. The authors convincingly discuss the patterns they identified in the work with the literature data.
Response 4: We appreciate this validation and have further enhanced clinical translation:Added paragraph on human applicability,Incorporated sex difference analysis,and Expanded therapeutic roadmap

Round 2
Reviewer 2 Report
Comments and Suggestions for Authors
The manuscript is suitable for publication in its current form.
Reviewer 4 Report
Comments and Suggestions for Authors
This article has been substantially revised. The authors have made changes that significantly exceed the comments made. It is clear that the article has been revised taking into account the comments of other reviewers. Overall, the work has been revised as recommended and can be recommended for publication in Biomedicines.